# Ceftazidime–Avibactam for the Treatment of Multidrug-Resistant Pathogens: A Retrospective, Single Center Study

**DOI:** 10.3390/antibiotics11030321

**Published:** 2022-02-28

**Authors:** Maria Di Pietrantonio, Lucia Brescini, Jennifer Candi, Morroni Gianluca, Francesco Pallotta, Sara Mazzanti, Paolo Mantini, Bianca Candelaresi, Silvia Olivieri, Francesco Ginevri, Giulia Cesaretti, Sefora Castelletti, Emanuele Cocci, Rosaria G. Polo, Elisabetta Cerutti, Oriana Simonetti, Oscar Cirioni, Marcello Tavio, Andrea Giacometti, Francesco Barchiesi

**Affiliations:** 1Infectious Diseases Clinic, Ospedali Riuniti Umberto I, Via Conca 71, 60126 Ancona, Torrette, Italy; maria.dipietrantonio@ospedaliriuniti.marche.it (M.D.P.); pallottafrancesco1993@gmail.com (F.P.); sara.mazzanti@hotmail.it (S.M.); paolomantini90@gmail.com (P.M.); b.candelaresi@gmail.com (B.C.); silviaolivieri1992@gmail.com (S.O.); fginevri@hotmail.it (F.G.); giuliacesaretti@hotmail.it (G.C.); o.cirioni@univpm.it (O.C.); andrea.giacometti@ospedaliriuniti.marche.it (A.G.); 2Department of Biomedical Sciences and Public Health, Polytechnic University of Marche Medical School, Via Tronto 10/a, 60020 Ancona, Torrette, Italy; g.morroni@pm.univpm.it (M.G.); f.barchiesi@staff.univpm.it (F.B.); 3Faculty of Medicine, Polytechnic University of Marche Medical School, Via Tronto 10/a, 60126 Ancona, Torrette, Italy; jennifer.candi@gmail.com; 4Infectious Diseases, Ospedali Riuniti Umberto I, Via Conca 71, 60126 Ancona, Torrette, Italy; seforacastelletti@gmail.com (S.C.); marcello.tavio@ospedaliriuniti.marche.it (M.T.); 5Hospital Pharmacy, Ospedali Riuniti Umberto I, Via Conca 71, 60126 Ancona, Torrette, Italy; emanuele.cocci@ospedaliriuniti.marche.it (E.C.); rosariagerarda.polo@ospedaliriuniti.marche.it (R.G.P.); 6Anesthesia and Transplant Surgical Intensive Care Unit, Ospedali Riuniti Umberto I, Via Conca 71, 60126 Ancona, Torrette, Italy; elisabetta.cerutti@ospedaliriuniti.marche.it; 7Clinic of Dermatology, Department of Clinical and Molecular Sciences, Polytechnic University of Marche, Via Conca 71, 60126 Ancona, Torrete, Italy; o.simonetti@univpm.it; 8Infectious Diseases Unit, Azienda Ospedaliera Ospedali Riuniti Marche Nord, 61122 Pesaro, Pesaro and Urbino, Italy

**Keywords:** Gram-negative pathogens, multidrug resistance, ceftazidime–avibactam

## Abstract

Background: Ceftazidime/avibactam is a new cephalosporin/beta-lactamase inhibitor combination approved in 2015 by the FDA for the treatment of complicated intra-abdominal and urinary tract infection, hospital-acquired pneumoniae and Gram-negative infections with limited treatment options. Methods: In this retrospective study, we evaluate the efficacy of ceftazidime/avibactam treatment in 81 patients with Gram-negative infection treated in our center from January 2018 to December 2019. The outcome evaluated was 30-days survival or relapse of infection after the first positive blood culture. Results: the majority of patients were 56 male (69%), with median age of 67. Charlson’s Comorbidity Index was >3 in 58 patients. In total, 46% of the patients were admitted into the medical unit, 41% in the ICU, and 14% in the surgical ward. Of the patients, 78% had nosocomial infections, and 22% had healthcare-related infections. The clinical failure rate was 35%: 13 patients died within 30 days from the onset of infection. The outcome was influenced by the clinical condition of the patients: solid organ transplantation (*p* = 0.003) emerged as an independent predictor of mortality; non-survival patients most frequently had pneumonia (*p* = 0.009) or mechanical ventilation (*p* = 0.049). Conclusion: Ceftazidime–avibactam showed high efficacy in infections caused by MDR Gram-negative pathogens with limited therapeutic options.

## 1. Introduction

In 2017, the WHO published a list of antibiotic-resistant priority pathogens, a catalogue of 12 species of bacteria that pose the greatest threat to human health: the list is divided in 3 categories (critical, high and medium priority) according to the urgency of the need for new antibiotics.

The critical priority category includes multidrug-resistant Gram-negative bacteria (*Acinetobacter baumannii*, *Pseudomonas aeruginosa*, and various *Enterobacteriaceae*), especially *E. coli* and *K. pneumoniae* which are the most involved species in blood stream infections (BSIs) and a cause of concern due to the wide antibiotic resistance patterns [1,2].

Combination therapy seems to be more successful than monotherapy for the treatment of MDR Gram-negative infections (i.e., colistin–polimixin B or tigecycline in combination with a carbapenem) and could reduce the insurgence of antibiotic resistance. Indeed, colistin should be used in combination therapy to avoid the selection of resistant strains (CRE). New therapeutic options include the β-lactam–β-lactamase inhibitor combination ceftazidime–avibactam (CZA), used in monotherapy or combination with aztreonam [3]. CZA couples a well-known cephalosporin with a novel non-β-lactam β-lactamase inhibitor. Avibactam inhibits ESBLs, AmpC β-lactamases (expressed in *Pseudomonas aeruginosa* and Enterobacteriaceae), class A carbapenemases (including the *Klebsiella pneumoniae* carbapenemase KPC) and OXA-48 β-lactamase family [4,5].

CZA is indicated for the treatment of complicated intra-abdominal infections (cIAI), complicated urinary tract infections (cUTI), hospital-acquired pneumonia (HAP) including VAP, and infections due to Gram-negative organisms with limited treatment options. The recommended intravenous dose in adults with creatinine clearance >50 is 2 g every 8 h [6,7,8,9]. Recent studies demonstrated that CZA was a promising drug for the treatment of severe KPC-producing *K. pneumoniae* (KPC-Kp) and reduced the mortality in BSIs patients [10,11,12].

The aim of this retrospective, observational study was to evaluate the efficacy of CZA administered in Gram-negative infections at a university hospital located in Central Italy.

## 2. Results

During the study period, a total of 81 patients received CZA therapy. Baseline characteristics of the patients included in the study are shown in Table 1. The majority were male (69%) with a median age of 67 years. Ninety-four percent of patients presented comorbidity, the most frequent being cardiovascular, renal and neurological diseases (67%, 30% and 28%, respectively). The Median Charlson Comorbidity Index was 5.

Forty-six percent of patients were hospitalized in medical wards and forty-one percent in ICUs. Septic shock was present in 20% of the overall population and pneumoniae in 82% of patients. Fourteen percent of pneumoniae cases were ventilator associated. A high proportion of patients carried a central venous catheter (CVC) (68%) and urinary catheter (CV) (68%). Solid organ transplantation (SOT) was the most common type of surgery characterizing these patients (28% of patients who had undergone surgery). In our case, liver transplantation was the only type of SOT.

CZA was mainly prescribed in complicated infections with limited therapeutic options (46%). The other cases of prescription included HAP/VAP (30%), cUTI (15%) and cIAI (9%).

The most frequently isolated pathogens were *K. pneumoniae* in 79% of cases, *P. aeruginosa* in 12%, and *E. coli* in 6%. Other pathogens were isolated in 2% of the cases, while in 7% of the cases, CZA was administered as an empirical treatment. Thirty-eight percent of patients had mixed infection. The majority of *K. pneumoniae* strains were KPC producers (95%). All strains were susceptible to CZA. The sites of isolation were bronchial secretions or pleural fluid (33%), blood (28%), urinary tract (19%), wounds (7%) and intra-abdominal fluid (6%).

CZA was administered with other antibiotics in 62% of patients: 19% of cases with tigecycline, 16% with colistin, 12% with fosfomycin, 12% with gentamicin, 11% with meropenem and 4% with amikacin. The median time of ceftazidime–avibactam administration was 11 days (Figure 1). Moreover, 29 patients had received other antibiotics in the 30 days before the administration of CZA.

Sixty-four percent of patients achieved a successful outcome, while thirty-six percent of patients did not. Of these, 13 patients died within 30 days from the onset of infection (30-day crude mortality 16%), while 16 patients presented an infection relapse (microbiological failure rate 21%). Among them, 12 patients survived, while 4 died. The 12 surviving patients with infection relapse were treated with CZA (38%) or with other therapy (62%).

A significantly higher proportion of patients with clinical failure received SOT (*p* = 0.003), mechanical ventilation (*p* = 0.049), or had pneumoniae (*p* = 0.009). Conversely, patients with successful clinical outcome were hospitalized more frequently in surgery wards (*p* = 0.04). No statistically significant differences were observed in treatment-related variables.

In the multivariate logistic regression analysis, only SOT emerged as independent predictors of failure treatment [OR 12.100 (1.369–106.971), *p* = 0.025].

## 3. Discussion

Infections caused by multidrug-resistant Gram-negative germs represent a major cause of mortality and a challenge for the physician. CZA is a new cephalosporin/beta-lactamase inhibitor combination approved in 2015 by the FDA for the treatment of complicated intra-abdominal and urinary tract infection, hospital-acquired pneumoniae and Gram-negative infections with limited treatment options. In this study, we retrospectively evaluated the Gram-negative bacterial infections treated with CZA that occurred in the Ospedali Riuniti Umberto I Hospital, in the period between January 2018 and December 2019. The purpose of the study was to evaluate the efficacy of CZA and risk factors related to 30-day mortality in subjects treated with this antibiotic.

The clinical success achieved in patients treated with ceftazidime–avibactam was 64%. In the literature, the success rate of the treatment ranges from 53% to 71% [10,11,12]. The variability is related to the different populations enrolled in the studies, the different species isolated, the site of the infections and the criteria used.

In our study, the infections treated with CZA were caused by several Gram-negative pathogens: the most isolated species was *K. pneumoniae* (79%), followed by *P. aeruginosa* (12%) and *E. coli* (6%). Uncommon isolated Gram-negative were *Stenotrophomonas maltophilia* (1%) and *Klebsiella aerogenes* (1%). Furthermore, in 38% of cases, the patients presented with polymicrobial infections. We observed a high relapse rate (21%). This percentage was higher than the data seen in the literature, in which relapses of BSI due to KPC-Kp is around 8–9% [10], while lower percentages were observed in infections other than BSIs (about 3–4%) [12]. Conversely, we observed that BSIs had a lower relapse rate (20%), compared to other infections (cUTI 25%, cIAI 50% and HAP/VAP 29%). Possible explanations for the high number of relapses may be the heterogeneity of the therapies administered, the delay of start therapy with CZA, or the non-homogeneity of the pathogens isolated. In the clinical practice, antibiotic therapy is often remodeled according to the patient’s clinical progress: worsening during the use of a therapeutic plan leads to a change in the chosen molecules, even if the duration of therapy is still shorter than that recommended. Additionally, therapy is often set empirically without waiting for the species identification and antibiotics susceptibility results.

According to other data described in the literature [10,12,13], our study demonstrated that mechanical ventilation and pneumonia were correlated with higher 30-day mortality. Our results agree with one retrospective observational multicentric Italian study including 138 adults with KPC-Kp infections who received CZA salvage therapy. The authors compared the 30-days mortality in 104 patients with KPC-Kp bacteremia who received CZA and 104 patients with KPC-Kp BSIs that were managed with regimens excluding CZA. In a multivariate analysis, mechanical ventilation resulted in being statistically associated with 30-days mortality [10]. In another retrospective observational study recently published, pneumonia was a variable independently associated with 14-days mortality in 47 patients treated for >72 h with CZA for KPC-Kp infections [13].

In our study, surgical patients showed greater clinical success with CZA therapy. In fact, in the most cases, these patients had a surgical site infection, a less serious clinical condition than patients admitted to medical and intensive care wards, and often underwent surgery to control the source of infection.

Interestingly, the only variable independently associated with the failure of CZA therapy at multivariate analysis was SOT. Carbapenem-resistant *K. pneumoniae* infection was an independent risk factor for mortality in liver transplantation recipients in some study [14,15,16] and is more frequent in these patients than in the general population, ranging from 6% to 23% [17,18,19]. Initially, few data were available in literature about the efficacy of CZA in patients with liver transplantations. In a recent study, Chen et al. evaluated the efficacy and safety of CZA in 21 patients infected by carbapenem-resistant *K. pneumoniae* after liver transplantation [20]. The 14-day and 30-day mortality rates were 28.6% and 38.1%, respectively, consistent with other reports [10]. The fact that SOT represented an independent risk factor for a negative outcome can be caused by the frequent surgical complications, long hospitalization and polymicrobial infections observed in this subgroup of patients.

## 4. Methods

The setting was the 980-bed Regional University Hospital in Ancona, tertiary referral center. A cohort of 81 patients, treated with at least 72 h of CZA therapy and who were ≥18 years old, with a Gram-negative infection diagnosed between January 2018 and December 2019, was considered: 46% of them were admitted in the medical unit, 41% in the ICU, and 14% in surgical wards.

Patient variables included age, sex, presence of acute or chronic comorbidities (i.e., diabetes, COPD, cancer, chronic hepatitis, chronic kidney disease, HIV, neutropenia, and solid organ transplantation), Charlson’s Comorbidity Index [21]) and APACHE II score, previous surgery, steroid and/or immunosuppressive therapy (≤30 days before BSI onset), and any invasive procedures (≤72 h before BSI onset). The isolation of KPC strains from other sites (≤30 days) or concomitant infections were also considered. Sepsis or septic shock were evaluated according to the criteria of the International Consensus Definition for Sepsis and Septic shock [22]. Hospitalization variables included nosocomial or healthcare-related infection, days between admission and onset of infection, and total days of hospitalization in the previous year. Treatment variables included antibiotic therapy with ceftazidime–avibactam in monotherapy or combination therapy, antibiogram availability, type and number of drugs, the use of ceftazidime–avibactam as salvage therapy after first-line treatment with other antimicrobials or in first-line therapy [10].

The outcome measured was death, relapse or persistence of infection within 30 days from the first positive blood culture.

The identification of species was performed with MALDI-TOF mass spectrometry (bio-Merieux, Marcy l’Etoile, France), and the detection of KPC was assessed with Genexpert (Cepheid, Sunnyvale, CA, USA). Susceptibility testing was performed by Vitek 2 system (bio-Merieux, Marcy l’Etoile, France) and interpreted according to the EUCAST 2022 definition [23], excluding ceftazidime–avibactam susceptibility, which was determined by MIC Test Strip (Liofilchem, Roseto degli Abruzzi, Italy).

Categorical variables were expressed as absolute numbers and their relative frequencies and compared by the χ^2^ or Fisher exact test; continuous variables were expressed as median and interquartile range (IQR) and evaluated by the Wilcoxon test and the Mann–Whitney U test (for no normally distributed variables). Variables that reached a statistical significance (*p* < 0.05) at univariate analysis were analyzed by multivariate logistic regression analysis to identify independent risk factors for mortality. The results obtained were analyzed using the software package SPSS 20.0 (IBM, Armonk, NY, USA).

## 5. Conclusions

In conclusion, CZA showed high efficacy in infections caused by MDR Gram-negative pathogens with limited therapeutic options. This study has some limitations related to its single center, retrospective nature, the statistical heterogeneity, the limited number of patients included in the analysis, the heterogeneity of the isolates and the therapies. Additional data produced by clinical practice are needed to elucidate the role of this molecule in managing infections caused by Gram-negative pathogens with limited therapeutic options.

## Figures and Tables

**Figure 1 antibiotics-11-00321-f001:**
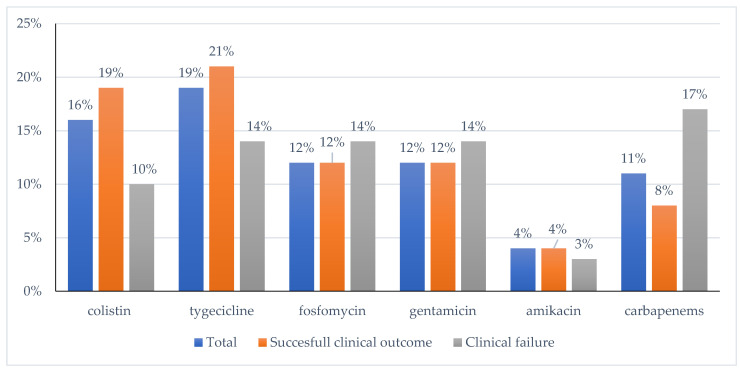
Clinical results of administration of CZA in combination therapy. The percentage showed refers to the total number of patients for each group (total, successful clinical outcome, and clinical failure).

**Table 1 antibiotics-11-00321-t001:** Demographic and clinical characteristics of the study cohort.

	All (*n* = 81)	Successful Clinical Outcome (*n* = 52)	Clinical Failure (*n* = 29)	*p*
**Variables**				
Patients variables				
Sex				
Male	56 (69%)	39 (75%)	17 (59%)	0.126
Female	25 (31%)	13 (25%)	12 (41%)	
Age (years) mean (IQR)	67 (56–75)	67 (55.75–75.25)	67 (58–75)	1
Charlson’s Comorbidity Index ≥ 3	58 (73%)	34 (67%)	24 (83%)	0.121
Comorbidities				
Diabetes	17 (21%)	9 (17%)	8(27%)	0.276
COPD	7 (9%)	4 (8%)	3 (10%)	0.697
Hematological malignancies	11 (4%)	7 (14%)	4 (14%)	1
Solid tumors	17 (21%)	9 (17%)	8 (28%)	0.278
Chronic Hepatitis	15 (19%)	9 (17%)	6 (21%)	0.707
Cardiovascular Disease	54 (67%)	37 (71%)	17 (59%)	0.251
Neurological disease	22 (28%)	13 (25%)	9 (32%)	0.495
Chronic kidney disease	24 (30%)	14 (27%)	10 (35%)	1
HIV	2(3%)	1 (2%)	1 (3%)	1
Neutropenia	2(3%)	1 (2%)	1 (3%)	1
Gastrointestinal disease	15 (19%)	9 (17%)	6 (21%)	0.707
SOT	8 (10%)	1 (2%)	7 (24%)	**0.003**
Wards submitting index culture				
Intensive care unit	33 (41%)	19 (37%)	14 (48%)	0.320
Surgery	11 (14%)	10 (19%)	1 (3.4%)	**0.04**
Medicine	37 (46%)	23 (44%)	14 (48%)	0.726
Pre-infection variables				
Central venous catheter	55 (68%)	34 (64%)	21 (72%)	0.223
Nasogastric tube	5 (6%)	2 (4%)	3 (10%)	0.244
Surgical drainage	13 (16%)	8 (15%)	5 (17%)	1
Bladder catheter	55 (68%)	34 (65%)	21 (72%)	0.516
Endoscopy ^a^	4 (5%)	3 (6%)	1 (3%)	1
Mechanical ventilation ^a^	11 (14%)	4 (8%)	7 (24%)	**0.049**
CVVH	13 (16%)	6 (12%)	7 (24%)	0.206
Steroid therapy ^b^	27 (33%)	17 (33%)	10 (35%)	0.870
Immunosuppressive therapy ^b,c^	14 (17%)	6 (12%)	8 (28%)	0.067
Previous surgery ^d^	43 (53%)	30 (58%)	13 (45%)	0.266
Infection variables				
Nosocomial infection	63 (78%)	40 (77%)	23 (80%)	0.804
Polymicrobial infections	31 (38%)	22 (42%)	9 (31%)	0.317
Septic shock	16 (20%)	11 (21%)	5 (17%)	0.672
Pneumoniae	66 (82%)	38 (73%)	28 (97%)	**0.009**
Sites of isolation				
Urinary tract	15 (19%)	10 (19%)	5 (17%)	0.825
Bronchial/pleural fluid	27 (33%)	16 (31%)	11 (38%)	0.512
abdominal fluid	5 (6%)	2 (4%)	3 (10%)	0.343
wounds	6 (7%)	6 (12%)	0	0.083
blood	23 (28%)	16 (31%)	7 (24%)	0.526
Pathogens				
*K. pneumoniae KPC*	64 (79%)	22 (76%)	32 (62%	0.225
*P. aeruginosa*	10 (12%)	4 (14%)	6 (12%)	0.739
*E. coli*	5 (6%)	3 (10%)	2 (4%)	0.534
Other ^e^	2 (2%)	0	2 (4%)	0.534
Empirical use	6 (7%)	0	6 (12%)	0.08
Treatment variables				
Previous therapy with others regimens ^b,f^	29 (36%)	18 (35%)	11 (38%)	0.729
Days of antibiotic therapy (median)	11 (7–14)	10 (7–14)	13 (7–14)	0.419
Combination therapy	50 (62%)	32 (62%)	18 (62%)	

Data are expressed as No. (%) unless otherwise specified. Abbreviations: IQR—interquartile range, COPD—chronic obstructive pulmonary disease, SOT—solid organ transplantation, CVVH—continuous veno-venous hemofiltration. ^a^ During the 72 h preceding BSI onset. ^b^ During the 30 days preceding BSI onset. ^c^ Excluding therapy with steroids. ^d^ During the 3 months preceding BSI onset. ^e^ Others: 1 *S. maltophilia* and 1 *K. aerogenes.*
^f^ Previous therapy: colistin plus tygecicline plus meropenem; 5 colistin plus meropenem; 4 colistin; 1 gentamicin plus tygecicline; 3 tygecicline plus meropenem; 2 gentamicin; 2 tygecicline; gentamicin plus colistin plus meropenem; 1 colistin plus tygecicline plus fosfomycin plus tygecicline; 2 cephalosporins plus fosfomycin; 1 cephalosporins; 1 quinolone; 2 quinolone plus meropenem; 1 ceftolozane–tazobactam; 2 meropenem.

## Data Availability

Data were collected from the medical case sheets and the laboratory and radiology data, available on the hospital’s electronic database.

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
