# Peer review of "Ceftazidime–Avibactam for the Treatment of Multidrug-Resistant Pathogens: A Retrospective, Single Center Study"

_antibiotics, 2022, doi:10.3390/antibiotics11030321_

Round 1
Reviewer 1 Report
In this manuscript, the authors performed a retrospective study to determine the efficacy of Ceftazidime/avibactam (CZA) treatment against gram-negative bacterial infections in patients. This observational study was performed on data gathered from 81 patients from a single center. These small sample size and single center analysis are limitations of the study, which the authors acknowledge. Overall, the analysis and conclusions presented by the authors are sound and scientifically the manuscript is fine. However, the general writing structure of the manuscript make it difficult to follow and read. In my opinion, I think the manuscript requires some restructuring and better explanations in some sections which would make it more impactful. Some specific and general comments are listed below, which should be rectified throughout the manuscript. I recommend the manuscript be accepted after the general comments are addressed.
General Comments
- Lines 39-64: The introduction section requires some restructuring. There are too many one sentence paragraphs, which doesn’t flow very well and seems like a bunch of bullet points put together. Since this study looks at CZA usage, a bit more in depth information about CZA in the introduction seems warranted.
- Line 73: Please correct ‘was’ to ‘were.’
- Lines 80-83: ‘CZA was prescribed….in 54%.’ Please correct the sentence structure and simplify the explanation.
- Lines 86-87: ‘Empirical treatment….other pathogens.’ The two parts of the sentence do not give complementary information. Please clarify this sentence.
- Lines 89-90: Please remove one sentence paragraph here and throughout the manuscript.
- Figure 1 is poorly explained. Moreover, the figure legend is poorly written.
- Line 100: Please correct ‘survival’ to ‘surviving.’
- Line 135: Please correct ’from to 53% at 71%’ to ’53% to 71%.’
- Line 139-140: “K. pneumoniae….1% of these.’ Please correct sentence structure.
- The methods section requires some restructuring and better explanation. Please see comment 1.
Reviewer 2 Report
Dear Authors
Thank you very much for your manuscript submission. Indeed, your work is well-designed and represents invaluable data. However, some revisions are needed.
- There are many typos and grammatical errors e.g.: Gram-negative/Gram-positive not gram-negative/gram-positive
- Please use more graphs in your manuscript. The manuscript is compacted by a mass of numbers and percentages. The use of more graphs softens the manuscript for the readers.
- I recommend two papers which supports your manuscript to have fruitful Introduction and Discussion sections. Please read and add the following papers to the References sections of your manuscript.
Antimicrobial Agents and Urinary Tract Infections. Curr Pharm Des. 2019;25(12):1409-1423. doi: 10.2174/1381612825999190619130216. PMID: 31218955.
Metallo-ß-lactamases: a review. Mol Biol Rep. 2020 Aug;47(8):6281-6294. doi: 10.1007/s11033-020-05651-9. Epub 2020 Jul 11. PMID: 32654052. - In Methods section you have mentioned:
"dentification of species was performed with MALDITOF mass spectrometry", please give a reference. Why did you choose this method. Please add the related figure to your manuscript.
"Detection of KPC was assessed with Genexpert".
Please add the related figure to your manuscript.
" Antibiotic MICs were determined by the reference broth microdilution method and results were interpreted according to the EUCAST definition". Please add the year and the version of applied EUCAST within the text.
"Ceftazidime-avibactam susceptibility was determined with Etest MIC Test Strip". Please add the related figure to your manuscript. - Please interpret the statistical calculations in details in Discussion section.
- I recommend you to add a flow chart or schematic figure to your manuscript to show the recruited procedures at one glance to the readers.
Round 2
Reviewer 2 Report
Accept